# Effect of Antioxidants on Thermo-Oxidative Stability and Aging of Bio-Based PA56T and Fast Characterization of Anti-Oxidation Performance

**DOI:** 10.3390/polym14071280

**Published:** 2022-03-22

**Authors:** Qiang Xu, Bing Guan, Weihong Guo, Xiucai Liu

**Affiliations:** 1Institute of Bio-Based Materials, East China University of Science and Technology, Shanghai 200237, China; xuqiang00721@163.com (Q.X.); guoweihong@ecust.edu.cn (W.G.); 2Shanghai Cathay Biotechnology Co., Ltd., Shanghai 200237, China; guanbing@cathaybiotech.com

**Keywords:** bio-based polyamide, thermal oxidation aging, rotational rheometer, DMA

## Abstract

Bio-based polyamide 56T (PA56T) is a new type of bio-based polyamide regarded as a promising material for sustainable solutions. The stabilization of PA56T compounded with Irganox 1098, Doverphos S9228, or SH3368 was studied by using a rotational rheometer and a circulating air oven at 150 °C. The thermal-oxidative aging resulted in an increase of the yellow color index of the PA56T/GF composites, which due to the carbonyl group as a chromophore group, continuously formatted during the aging process. After 10 days of aging, the mechanical properties and dynamic mechanical properties increase due to the molecular cross-linking and annealing effects. When the aging time is beyond 20 days, the degradation of molecular chain segments dominates, and the mechanical properties of PA56T/GF deteriorate continuously. The addition of antioxidants only slowed this effect and did not change the process of thermal-oxidative aging, which destroys the molecular chain. The results from both methods are consistent after a series of characterizations by FTIR, XRD, and so on. In the case of samples without lubricant, the rotational rheometer has the benefit of being less time-consuming than the accelerated aging experiment.

## 1. Introduction

Polyamide (PA) is one of the most promising thermoplastic engineering plastics due to its inherent properties [1]. In most of the polyamides made from fossil fuels, carbon end up as carbon dioxide in the atmosphere and contributes to global warming [2]. Bio-based polyamide, as an alternative approach, has provided an opportunity to reduce carbon emissions for this polymer family. Currently, only a few polyamides can be partially or totally replaced from biosources [3]. Polyamide 56 (PA56), based on 1, 5-pentane diamine from biomass, is one with excellent fluidity, moisture absorption, and mechanical properties for applications in the textile industry [4], but the applications of PA56 in other areas are frequently limited by its high moisture absorption rate and low thermal stability compared with PA66. By introducing terephthalic acid into the polymer chains, polyamide 56T (PA56T) has a significant improvement in moisture absorption and mechanical properties, which is desirable for application as a thermoplastic engineering plastic. 

However, after thermal aging, the chain end amino group of PA56T is still easily oxidized and causes yellowing and degradation, leading to a performance loss of the modulus, and ultimately the failure of the product performance [5,6,7,8]. Based on the thermal-oxidative aging mechanism of polyamides, the problem of thermal aging is usually solved by adding stabilizers, such as hindered phenols, phosphite ester, and copper salts [9,10,11]. Previously published papers have elucidated the mechanism of stabilizers on polyamides [12,13,14,15,16]. For determining the effects of stabilizers, the accelerated aging experiment is generally carried out in an incubator with certain temperatures, which takes one to two months [17,18]. In addition to this method, Qian Zhijun determined thermal stability for nylon using a torque rheometer to shorten the time spent on this characterization [19]. Thermal analysis is also reported to be particularly simple and very often used to study polyamide degradation [20,21], which determines the efficiency of stabilizers as well as oxidation induction time (OIT) [22,23]. However, OIT is very dependent on the experimental conditions (heating rate, sample size, etc.) [24]. Recently, several kinds of literature have reported that the rotational rheometer had been used to characterize the molecular structure well in the field of polyamide research, such as with PA66 and PA6 [25,26]. Therefore, an attempt was made here to characterize the changes of the molecular structure of PA56T in oxidation by using a rotational rheometer and then assess the thermal stability of PA56T.

In this paper, compound Irganox 1098 was used to improve the thermal stability of PA56T due to its excellent compatibility with polyamide. Considering the high processing temperature (290 °C), compounds S9228 and SH3368 were also used for PA stabilization because of their high-temperature resistance. Here, we investigated the thermal stability of PA56T with different stabilizers, in which a rotational rheometer was used to assess the stability of PA56T with/without antioxidants during accelerated thermal aging experiments.

## 2. Experimental

### 2.1. Materials

Bio-based PA56T was supplied by Cathay Biotechnology Co., Ltd. Glass fiber with a length of 3 mm was provided by Taishan Glass Fiber Co., Ltd. (Taian, China). The antioxidants studied were Irganox1098 (From BASF, Shanghai, China) and S9228 (From Doverphos Company, Shanghai, China), with chemical structures shown in Figure 1. The PA56T sample has a ratio of 2:3 of adipic acid: terephthalic acid and a melting temperature of 269 °C. The SH3386 is a synergistic mixture of an antioxidant of copper compound, organic amine copper complex, and effective hydrolysis-resistant lubricant.

### 2.2. Preparation of PA56T Samples

Antioxidants, glass fibers, and PA56T were blended by a co-rotating twin-screw extruder (*L/D* = 40, *D* = 26 mm, TDS-26B, Nanjing Noda Company, Nanjing, China). The temperature profiles from the feeding zone to the die were set to 180 °C/260 °C/280 °C/290 °C/290 °C/290 °C/290 °C/290 °C and the screw speed was set to 400 rpm. The pellets of PA56T were dried in a vacuum oven at 105 °C for 6 h. The specimens were then produced via injection molding (MA900II/260, Haitian Plastic Machinery Company, Ningbo, China). The temperature of each section from the feeding zone to the injection port were set to 265 °C/275 °C/290 °C/290 °C/290 °C. The injection pressure and storage backpressure were 70 Pa and 3 Pa, respectively. The retention time was 3 s. The mold temperature for injection molding was 80 °C. Tensile specimens, bending specimens, and notched impact specimens were prepared according to GB/T1040.2, GBT9341-2008, and GB/T1043.1, respectively. Table 1 is the ratios of each component. 

### 2.3. Aging

All PA56T samples were placed into a ventilated oven, at 150 °C for 50 days. The samples were left sealed at room temperature for one day before aging. After heating and at the given aged time, samples were removed and cooled in a sealed bag to avoid humidity absorption, and experiments were carried out immediately.

### 2.4. Characterization 

#### 2.4.1. Rotational Rheometer

A rotational rheometer (MARS 3, Thermo Hakke, Waltham, MA, USA) was used to test the viscosity of PA56T at an experiment temperature of 280 °C with a rotor speed of 0.01 r·s. The chamber was purged for three minutes with nitrogen before placing the pellets. The sample was placed into the rotor and the rotor spacing was set to 1 mm. The sample was then kept in a nitrogen atmosphere for 10 min to stabilize the temperature, and it was kept under nitrogen for an additional 20 min, then the gas path was switched from nitrogen to air. The viscosity changes were recorded.

#### 2.4.2. Fourier Transform Infrared Spectroscopy

The infrared spectra were recorded using an FTIR spectrometer (Nicolet is50, Thermo Fisher Scientific, Waltham, MA, USA). Infrared spectra of samples were obtained by the accumulation of 32 scans at a resolution of 4 cm^−1^, ranging from 4000 to 400 cm^−1^. A small quantity of powder scraped off from the test samples was mixed with KBr at a mass proportion of approximately 1/100, and the KBr discs were prepared by compressing in a tablet press.

#### 2.4.3. Yellow Index (YI)

The yellow index of PA56T was characterized by a spectrometer (Ci7600, Shanghai Kaide Color Management, Shanghai, China) in standard reflected light mode. The YIs of samples 4 and 6 were not tested due to color stains produced by the addition of copper-containing antioxidants. 

#### 2.4.4. Crystallization Analysis 

X-ray diffraction patterns were recorded using a Copper Target X-ray diffractometer (18KW/D/max2550VB/PC, Rigaku Electric, Tokyo, Japan) using Cu Kα (λ = 0.154178 nm) radiation, 18 KW power, and 450 mA current. The angular resolution was 0.02 degrees. The scanning range was 5–75°, and the scanning rate was set as 5° min^−1^. Results were registered in the in-situ mode with a computer, and X-ray diffractograms of samples were treated using the JADE 5.0 Software (Jade 5.0, Beijing Bona Technology Co., Beijing, China).

#### 2.4.5. Mechanical Properties

The tensile strength and bending strength of samples were determined by using an electronic universal testing machine (UTM4304, Shenzhen Sansi aspect Technology Co, Ltd., Shenzhen, China) according to GB/T1040.2 and GBT9341-2008, respectively. The notched impact strength of the samples was tested with a pendulum impact test machine (PTM1000, Shenzhen Sansi aspect Co., Ltd., Shenzhen, China) according to GB/T 1843-2008. At least 3 specimens of each formulation were tested, and the average was taken in each of the groups.

#### 2.4.6. Microscopy Investigation

Scanning electron microscopy (SEM) images were observed on an S-3400N (S-3400N, Hitachi Limited, Tokyo, Japan) instrument, at an accelerating voltage of 5 kV and 15 kV under a high vacuum. The surfaces and fracture cross-sections of both aged and unaged PA56T/GF samples were made electrically conductive by sputter coating with a gold layer before the examination.

#### 2.4.7. Dynamic Thermo-Mechanical Analysis (DMA)

A dynamic Thermo-mechanical analyzer (DMA242E, Germany Nerzsch Company, Bavaria, Germany) was used to test the storage modulus, the loss modulus, and tan δ of the samples. The samples were heated from −30 °C to 200 °C, with a heating rate of 3 °C/min. The surface of the sample was polished smooth before the test. The deformation mode was three-point bending mode. The frequency was thermal

## 3. Results and Discussion

### 3.1. Rotational Rheometer Analysis

The viscosity curves of the fresh samples under study are presented in Figure 2. It was observed that the viscosity of PA56T was almost unchanged within the first ten minutes under the nitrogen. However, the viscosity of PA56T first decreased and then increased in the air atmosphere, which resulted from the polyamide decomposition and the cross-linking between molecular chains, respectively; the same conditions and mechanisms have been proposed in other studies of aliphatic polyamides 66 and 6 [27,28]. Verdu [29], in a kinetic study of PA12, a long chain aliphatic polyamide, pointed out that two mechanisms for the generation of radicals are polymer thermolysis and hydroperoxides decomposition. This led to a severe cross-linking reaction in a short time, which caused a sharp rise in viscosity (see neat PA56T in Figure 2). As aging time increased, the effect of viscosity was gradually dominated by cross-linking reaction. Contrary to the non-stabilized sample 1, the viscosity of PA56T with stabilizers changed less. The different stabilizers understudy affected the thermal stability of PA56T to different degrees. Table 2 displays the results for the time of each stage and the apparent viscosities of samples in the test. 

During the compounding process, the degradation of PA56T was caused by high temperature and high shear stress. In Table 2, sample 5 had the highest apparent viscosity, meaning that it had the best thermal stability. However, the apparent viscosities of sample 4 and sample 6 are lower than that of sample 1, which may be caused by the lubricant in organic copper antioxidants. In the case of samples containing stabilizers, the duration of decrease of viscosity took longer than that of sample 1. It is well known that the oxidation process of polyamides was accompanied by decomposition and the cross-linking of molecular chains [30]. Based on the viscosity curves of samples, it appeared that their stabilization can be detected. The time of the decrease of viscosity will be longer, whereas the viscosity curve exhibits a migration to lower viscosity for stabilized samples. Therefore, it is concluded that sample 5, with both antioxidants of Irganox1098 and S9228, has the highest thermal stability under nitrogen atmosphere at 280 °C.

### 3.2. XRD Analysis

The X-ray diffraction patterns of the prepared samples, PA66 and PA56, are shown in Figure 3. PA66 has two diffraction peaks at 2*θ* = 20.7° and 2*θ* = 24.1°, which corresponds to the α1 crystal form and the α2 crystal form of PA66, respectively [31]. PA56T has a single diffraction peak that appears at 2*θ* = 21.5°, with *γ* phase as the main component, which is similar to PA56. The addition of the antioxidants did not alter the crystal form of PA56T, but the peak shape had changed (Figure 3b). Table 3 shows the grain size of samples according to the Scherrer formula: (1)$$D=\frac{K\gamma }{B\cos\theta }$$
where *K* is the Scherrer constant. *K* = 0.89 when B is the half-height width of the diffraction peak and *K* = 1 as B is the integral height and width of the diffraction peak. *D* is the average thickness of the crystal grain perpendicular to the crystal plane direction (nm), *B* is the half-height width of the measured sample diffraction peak, which needs to be converted into radians (rad) during the calculation process, *θ* is the Bragg diffraction angle, in degrees, and *γ* is the X-ray wavelength, which is 0.154178 nm. 

The results showed that the grain size of sample 1 was 1.44 nm, which was the largest grain size among all samples under this study. The grain sizes of samples 5, 3, and 2 were 1.06 nm, which is smaller than that of other samples. It is speculated that the high apparent viscosities of PA56T (Table 2) and difficulty in crystallization will result in small crystal grain sizes during the quenching process. Some literature on the crystallization of polyamides showed that, in the condition of high cooling rate (quenching process), the speed of the polyamide chains folding into the lattice cannot keep up with the cooling rate, with a corresponding reduction in the number and size of crystallization of polyamides [32,33]. For samples 4 and 6, the reason for low viscosity may be due to the presence of lubricant counteracting this negative influence. 

### 3.3. Groups and Yellow Index Analysis 

During aging, the FTIR spectra of PA56T are depicted in Figure 4, where a wide absorption at 1715 cm^−1^ gradually appears due to the generation of carbonyl groups [34]. The results reveal that PA56T with 1098 antioxidants showed high carbonyl content compared to PA56T with other antioxidants. The reason for this may be that the quinone compounds as the final product of hindered phenol antioxidant contain many carbonyl groups. This is corroborated by the study on the mechanism of phenolic antioxidants for polyamide 11 [35].

Figure 5 shows the reaction of phenolic antioxidants to destroy free radicals and the generation of quinone compounds. It can be seen in Figure 6 that all samples have different colors, and the yellow index would be used to indicate their level of yellowing.

The data showed that the compound antioxidants well inhibited the yellowing of sample 5 with a YI of 18.36, and the YI of sample 2 was higher than that of samples 3 and 5. It was speculated that the hydroperoxide did not decompose in time and promoted the oxidation of polyamide. It is well known that the production of hydroperoxide during oxidation of polyamide can be decomposed by phosphate antioxidants [36], which is not present in the case of PA56T stabilized by hindered phenols. The PA56T showed lower YI as compared to research for PA66 [37]. The PA56T, as a semi-aromatic nylon, exhibited higher thermal stability than the PA66 due to the introduction of the benzene ring on the main chain. Consequently, it was found that compound antioxidants reduced the production of carbonyl and increased YI (Figure 7), meaning that sample 5 showed the best stability in the FTIR and YI tests. 

### 3.4. Mechanical Properties

The changes in the mechanical properties of samples in accelerated aging experiments are presented in Figure 8. After the compounding process, the molecular weight of PA56T would decrease, which could be referred to as the apparent viscosity of each sample in Table 2 and led to the difference in their mechanical properties at day 0. By the third day of the aging experiment, it was observed that the tensile strength and flexural strength of sample 5 increased, while the notched impact strength decreased, which was mainly caused by the annealing effect. With the further increase of aging time, the mechanical properties of each sample decreased quickly, and then the downward trend gradually slowed down. By the fiftieth day of the aging experiment, the tensile strength and flexural strength of sample 5 were 184.6 MPa and 254.3 MPa, and the retention rates were 88.9% and 87.9%, respectively. The notched impact strength of sample 5 was 8.3 KJ/m^2^, and the retention rate was 78.2%, which was higher than the other samples. These results show that sample 5 maintained the highest mechanical properties, which were consistent with previous results in other aspects. 

### 3.5. Microscopy Investigation

Figure 9 shows the typical SEM images of the fracture surfaces of the PA56T/GF specimens. When the composites are under an external load, fiber breakage is a key to achieving a good mechanical performance compared to fiber pull-out. It can be seen that many glass fibers were broken and a polymer matrix was adhered to the surface of these fibers, while the fibers in the aged samples were pulled out of the matrix and dispersed irregularly. This indicated that the aging of the polymer matrix affected the interfacial properties with the glass fiber, further deteriorating the mechanical properties of the PA56T/GF samples. When the thermal-oxidative aging time was increased to 50 days, many smooth glass fibers and holes were exposed, and the mechanical properties deteriorated significantly, which was attributed to a decrease in the pullout forces provided by the glass fiber.

### 3.6. Dynamic Thermo-Mechanical Analysis (DMA) 

Figure 10 displays dynamic thermal-mechanical analysis curves as a function of temperature for the samples. In Figure 10a, it can be noticed that there was a remarkable decline in storage modulus in the temperature range 50–100 °C, which showed that the part of the molecular chain of PA56T gained mobility as the temperature approached the glass transition temperature. Evidently, the storage modulus of samples 2, 3, and 5 were higher than sample 1, which was because stabilizers maintain the rigidity of the PA56T during processing. The storage modulus of samples 4 and 6 were lower than that of sample 1 due to the increased migration of molecules by the addition of lubricant.

In Figure 10b, two distinctive peaks can be identified, α and β, which belong to the two relaxation processes in PA56T. The reason for the occurrence of β-relaxation is that the process of change from freezing to motion can occur for moving units smaller than the chain segments, although the chain segments of PA56T are frozen in the glassy state [38]. The instability of the energy storage modulus of PA56T/GF at low temperatures may be due to β-relaxation and water molecules, which some small moving units move and become another steady state. This phenomenon can also be seen in [39,40]. The strongest loss peak is the α relaxation, which indicates the beginning of relaxation motion in the chain segment of PA56T, according to polymer transition relaxation theory. 

The glass transition temperature (T_g_) of polyamide is always taken by the maximum of tan, δ, which implies a relaxation process and involves the extent of the mobility of small groups and the molecular chain segment. It was noted that the stabilizers increased the T_g_ of PA56T a little. However, the results of sample 4 and sample 6 were different; it was speculated that the lubricant may increase the mobility of the polymer molecules. 

The storage modulus, loss modulus, and Tg values of the PA56T/GF samples for different aging times at 150 °C can be seen in Figure 10. The average values of the energy storage modulus in the temperature range of 50–75 °C were recorded and used for comparison. The loss modulus and glass transition temperature were the maximum value of α-relaxation peak and the temperatures corresponding to tan δ_max_, respectively.

It was noted that the storage modulus of all the samples showed the same trend of increasing and then decreasing during the aging process (Figure 11a). The early increase in energy storage modulus may be attributed to polymer molecular chains and annealing effects during high-temperature aging. When the aging time was beyond 10 days, the decomposition of molecular chain segments gradually dominated, and then the energy storage modulus kept decreasing.

The changes of loss modulus of samples with aging time can be seen in Figure 11b. During aging, the loss modulus of PA56T decreased sharply in the early stage and tended to be flat during the later stage. This variation can be explained in light of two mechanisms: (1) micro-crystals can be viewed as a physical cross-linking effect where the increase of crystallinity of PA56T can reduce the creep and stress relaxation, (2) the weak molecular chains between the crystalline region and the crystalline region or between the crystalline region and the amorphous region break, which counteracts the effect of annealing. The study on the dynamic mechanical behavior of PA6 in oxidation also conforms to this rule [41].

In general, glass transition temperature (T_g_) is the transition temperature of a polymer from the glassy state to the highly elastic state. The higher the value of tan δ_max_, the greater is the degree of molecular mobility. The increase of T_g_ in aging time up to 10 days meant that the cross-linking of molecular chains and the increase of crystallinity reduce the mobility of molecules. This variation is in agreement with the results of storage modulus and tan δ. When the aging time arrived at 20 days, the storage modulus, loss modulus, and the value of tan δ would continue to decrease because the molecular degradation was predominant during the whole aging process, which could also be seen in [42] about PA10T.

## 4. Conclusions

This paper reports the effect of three antioxidants on the thermal stabilization of PA56T. The compound antioxidants well improved the stability of PA56T by retarding the carbonyl accumulation and molecular degradation. 

Compared to the accelerated aging test, the stability of samples can be measured well by the rotating rheometer tests. However, in the presence of lubricant, the results of a rotating rheometer appear to be less accurate with respect to stability. The result of XRD indicated that the crystal form of PA56T did not change after compounding with the antioxidants. The grain size changed due to the protective effect of the antioxidants on the molecular chains. FTIR results showed that the molecular chains were continuously oxidized with the increase of aging time, and the value of YI increased due to the formation of carbonyl products. The thermal-oxidative degradation of the PA56T/GF samples caused deterioration of the thermal stability as well as their mechanical properties. Therefore, all the results demonstrated that the crystallization property and the dynamic and static mechanical properties of PA56T/GF were dramatically influenced by thermal-oxidative aging time at 150 °C, and 20 days was the turning point in the article. 

All the results showed that the addition of antioxidants effectively slowed down the aging process of PA56T/GF, which could extend the service life of PA56T/GF. In addition, the most effective formulas can be identified in a short period of time by a rotating rheometer, which will save much time for application.

## Figures and Tables

**Figure 1 polymers-14-01280-f001:**
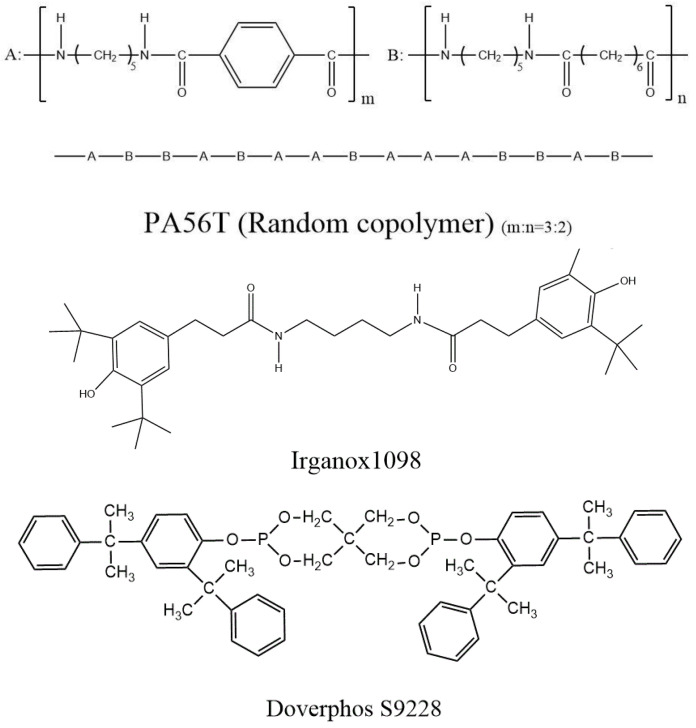
Chemical structure of PA56T, Irganox1098, and Doverphos S9228.

**Figure 2 polymers-14-01280-f002:**
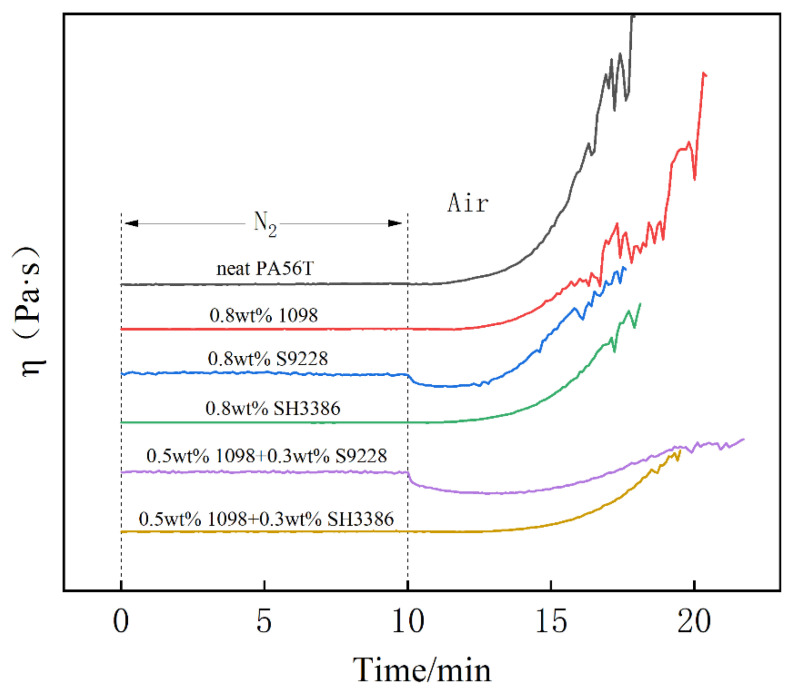
Viscosity curve of PA56T under nitrogen and air atmosphere at 280 °C.

**Figure 3 polymers-14-01280-f003:**
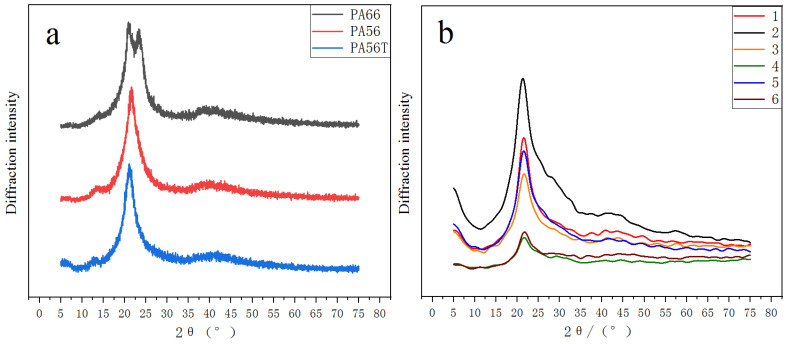
XRD diffraction pattern (**a**) XRD diffraction peaks of PA56, PA66, PA56T, (**b**) XRD diffraction peaks of samples with different antioxidants.

**Figure 4 polymers-14-01280-f004:**
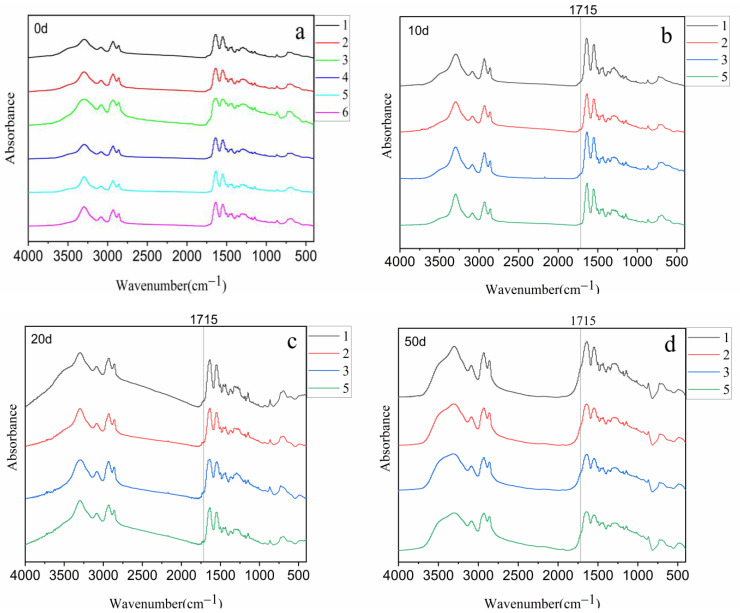
The infrared spectrum of samples: (**a**) sample before aging; (**b**) samples 1, 2, 3, and 5 after aging for 10 days; (**c**) samples of 1, 2, 3, and 5 after aging for 20 days; (**d**) samples 1, 2, 3, and 5 after aging for 50 days.

**Figure 5 polymers-14-01280-f005:**
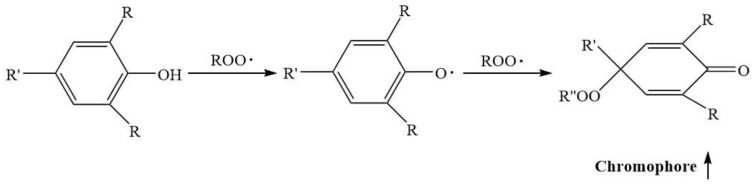
The reaction of phenolic antioxidants to destroy free radicals and the generation of chromogenic groups.

**Figure 6 polymers-14-01280-f006:**
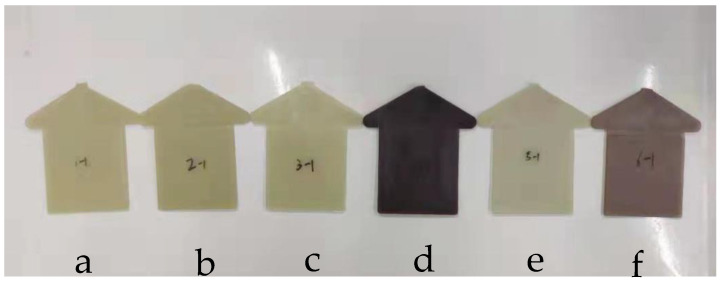
Comparison of the color of plate specimens at the beginning of aging (**a**) 1, (**b**) 2, (**c**) 3, (**d**) 4, (**e**) 5, (**f**) 6.

**Figure 7 polymers-14-01280-f007:**
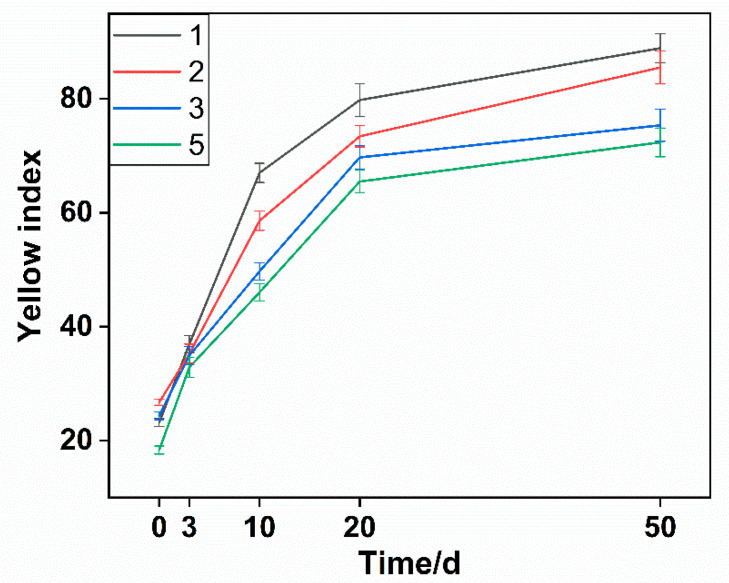
The influence of different antioxidants on the yellow index of the PA56T samples.

**Figure 8 polymers-14-01280-f008:**
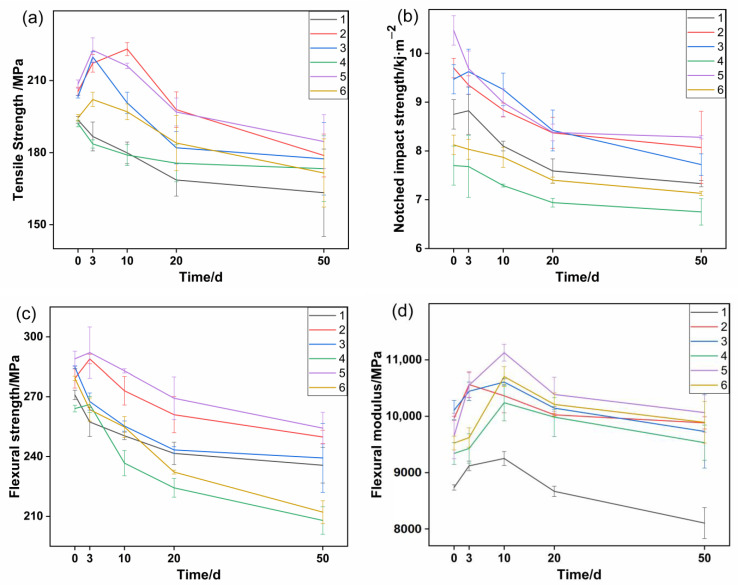
The effect of aging time on mechanical properties: (**a**) tensile strength, (**b**) notched impact strength, (**c**) flexural strength, (**d**) flexural modulus.

**Figure 9 polymers-14-01280-f009:**
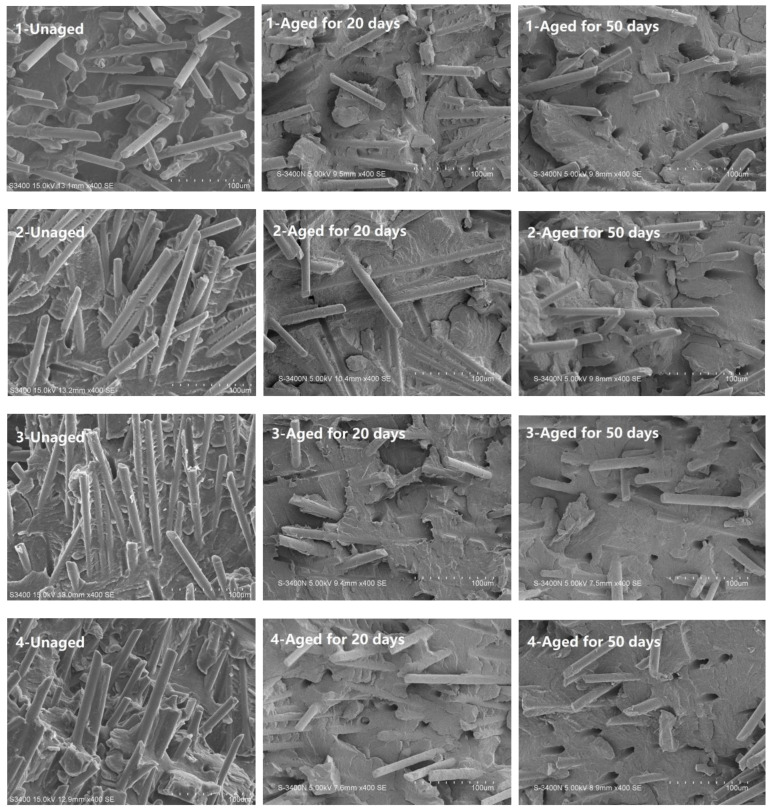
SEM images of the fracture surfaces of the unaged and aged PA56T/GF samples at 400× magnification.

**Figure 10 polymers-14-01280-f010:**
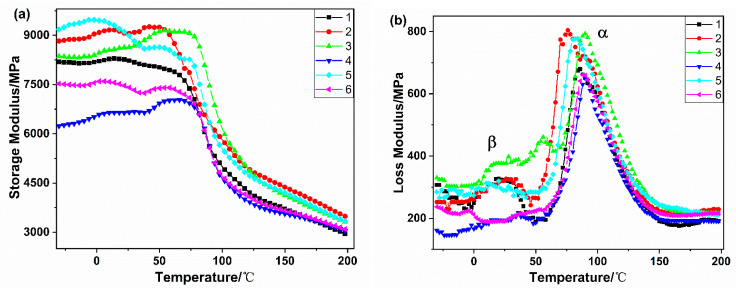
Dynamic thermo-mechanical analysis curve of PA56T/GF (**a**) storage modulus of the PA56T/GF samples; (**b**) loss modulus of the PA56T/GF samples; (**c**) tan δ of the PA56T/GF samples.

**Figure 11 polymers-14-01280-f011:**
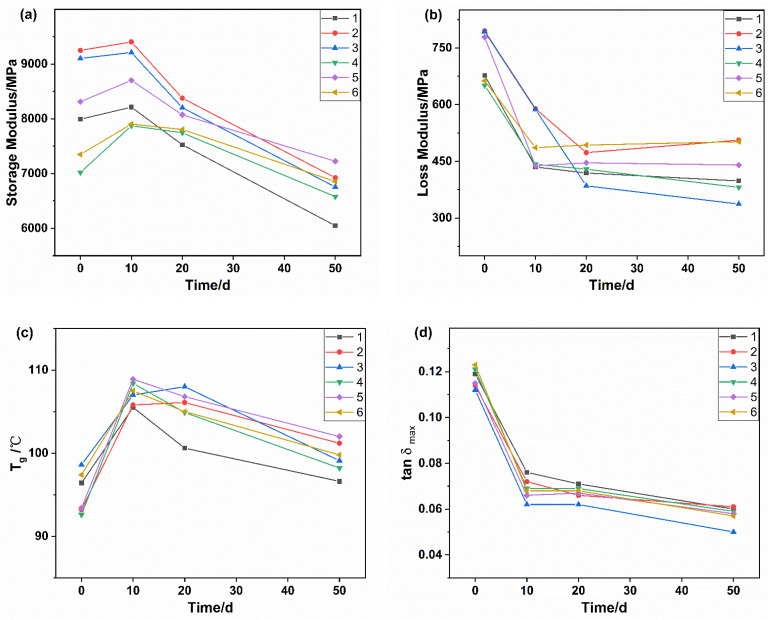
Storage modulus (**a**), loss modulus (**b**), Tg (**c**), and tan δ_max_ (**d**) of the PA56T/GF samples for different aging time at 150 °C.

**Table 1 polymers-14-01280-t001:** The formulas of samples.

Sample	PA56T (wt%)	Glass Fiber (wt%)	1098(wt%)	S9228 (wt%)	SH3386 (wt%)
1	67.0	33.0	0	0	0
2	66.2	33.0	0.8	0	0
3	66.2	33.0	0	0.8	0
4	66.2	33.0	0	0	0.8
5	66.2	33.0	0.5	0.3	0
6	66.2	33.0	0.5	0	0.3

**Table 2 polymers-14-01280-t002:** The apparent viscosity of PA56T with antioxidants, the minimum and maximum of viscosity, and the time of each stage (starting from 10 min).

Sample	Apparent Viscosity (Pa·s)	Time of Reaching the Valley (min)	Peak-Valley Value (Pa·s)	Time to Reach the Summit (min)	Peak Value (Pa·s)
1	5855	0.8	5493	2.1	193,100
2	6770	1.7	6070	10.5	182,000
3	19,933	1.8	11,500	7.9	92,700
4	4750	1.2	4610	8.6	86,900
5	21,500	2.8	10,400	11.7	47,900
6	5133	1.7	4833	9.5	61,390

**Table 3 polymers-14-01280-t003:** The grain size according to the Scherrer formula.

Sample	Half-Height (°)	Peak (°)	Grain Size (nm)
1	5.53	21.58	1.44
2	7.52	21.32	1.06
3	7.50	21.64	1.06
4	6.00	21.64	1.33
5	7.48	21.56	1.06
6	6.18	21.74	1.29

## Data Availability

Not applicable.

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
