# Peer review of "Effect of Antioxidants on Thermo-Oxidative Stability and Aging of Bio-Based PA56T and Fast Characterization of Anti-Oxidation Performance"

_polymers, 2022, doi:10.3390/polym14071280_

Round 1
Reviewer 1 Report
Please see the attachment file.

Author Response
请参阅附件。

Reviewer 2 Report
- In the Experimental methods, section 2.3, 2.4.4, can be improved.
- Conclusion can be improved.
Reviewer 3 Report
This manuscript presented a study about the effect of antioxidants on the properties of PA56T composites. The work need to be improved. Some results need to be deeply discussed, as pointed below.
Abstract: please be more specific about the results obtained in this work.
Section 2.2: What type of twin-screw extruder was used? A Corotating extruder? What was the temperature profile used?
Section 2,2: what conditions were used during injection molding (temperature, time, pressure)?
Table 1: Why the authors opted to used o.8 wt% of antioxidant additive?
Section 2.3: please add the total aging time.
Section 2.4.2: Were FTIR spectra obtained by ATR mode or KBr pellets?
Equation 4: the authors must describe the meaning of each variable in Scherrer equation.
Section 3.4: I suggest improve the discussion in this section. How the obtained results in this section can be correlated to the addition of stabilizers and the previous results of this work?
Section 3.5: improve the discussion in this section. Better discuss the effect of the addition of stabilizers and also the effect of aging on the dynamical mechanical properties. I suggest compare the Tg values and/or storage modulus (loss modulus) before and after aging and better discuss what can cause these differences.
Round 2
Reviewer 1 Report
The authors have modified carefully the manuscript as the reviewer's suggestions; hence, the manuscript can be accepted in its current form
Reviewer 3 Report
After corrections the manuscript reads well. I suggest publication in its current form.